# Practical Use of the (Observer)—Reporter—Interpreter—Manager—Expert ((O)RIME) Framework in Veterinary Clinical Teaching with a Clinical Example

**Amanda Nichole (Mandi) Carr** [1,2,3], **Roy Neville Kirkwood** [2] **and Kiro Risto Petrovski** [1,2,3,*]

1    Davies Livestock Research Centre, School of Animal and Veterinary Sciences, The University of Adelaide, Roseworthy, SA 5371, Australia
2    School of Animal and Veterinary Sciences, The University of Adelaide, Roseworthy, SA 5371, Australia
3    Australian Centre for Antimicrobial Resistance Ecology, School of Animal and Veterinary Sciences, The University of Adelaide, Roseworthy, SA 5371, Australia
*    Correspondence: kiro.petrovski@adelaide.edu.au

**Abstract:** This review explores the practical use of the (Observer)—Reporter—Interpreter—Manager—Expert ((O)RIME) model in the assessment of clinical reasoning skills and for the potential to provide effective feedback that can be used in clinical teaching of veterinary learners. For descriptive purposes, we will use the examples of bovine left displaced abomasum and apparently anestric cow. Bearing in mind that the primary purpose of effective clinical teaching is to prepare graduates for a successful career in clinical practice, all effort should be made to have veterinary learners, at graduation, achieve a minimum of Manager level competency in clinical encounters. Contrastingly, there is relatively scant literature concerning clinical teaching in veterinary medicine. There is even less literature available on strategies and frameworks for assessment that can be utilized in the different settings that the veterinary learners are exposed to during their education. Therefore, our intent for this review is to stimulate and/or facilitate discussion and/or research in this important area. The primary aim of preparing this review was to describe a teaching technique not currently used in the teaching of veterinary medicine, with potential to be useful.

**Keywords:** animal science; clinical activities; clinical practice; teaching; traditional academic setting; veterinary learners; work-based learning

## 1. Introduction

The aim of clinical teaching in veterinary medicine is to prepare new entrants into the profession to meet all required day-1 veterinary graduate competencies. One of the cornerstones in the development of veterinary learners and their transition into practitioners is the exposure to practice. Exposure to practice (experiential learning or work-based learning) aims to assist veterinary learners to develop veterinary medical and professional attributes within the specific clinical context of the work [1]. One problem with exposure to practice is the assessment of progress in the clinical (reasoning) competencies of learners. This may be even more difficult with the increasing number of veterinary schools opting for a partial or entirely distributed model of exposure to practice, as instructors in the distributed institutions may be inexperienced in the assessment of the progression in clinical competencies of learners. In this review, we provide a description of one assessment framework that can be utilized for this purpose.

Development of clinical competencies in veterinary learners is heavily dependent on effective feedback [2–7]. With the aim of enhancing the learner's performance, feedback must be relevant, specific, timely, thorough, given in a 'safe environment', and offered in a constructive manner using descriptive rather than evaluative language. For good execution

of provision of effective feedback, instructors should receive formal training in the delivery of feedback [2,8–12].

### 1.1. Basics of Assessment of Clinical Competency of Veterinary Learners

The foundation of assessment should be addressing domains stated within Miller's pyramid and its extension (Figure 1) [13–15]. Miller's pyramid gave rise to competency-based medical education [13,14]. A variety of frameworks for assessment of clinical competencies are available to medical instructors [14,16]. Detailed discussion of all frameworks is beyond the scope of this article. Readers are recommended to read reviews or appropriate articles on assessment of clinical competencies in learners in medical fields (e.g., [16,17]) and/or educational psychology related to veterinary learners [18].

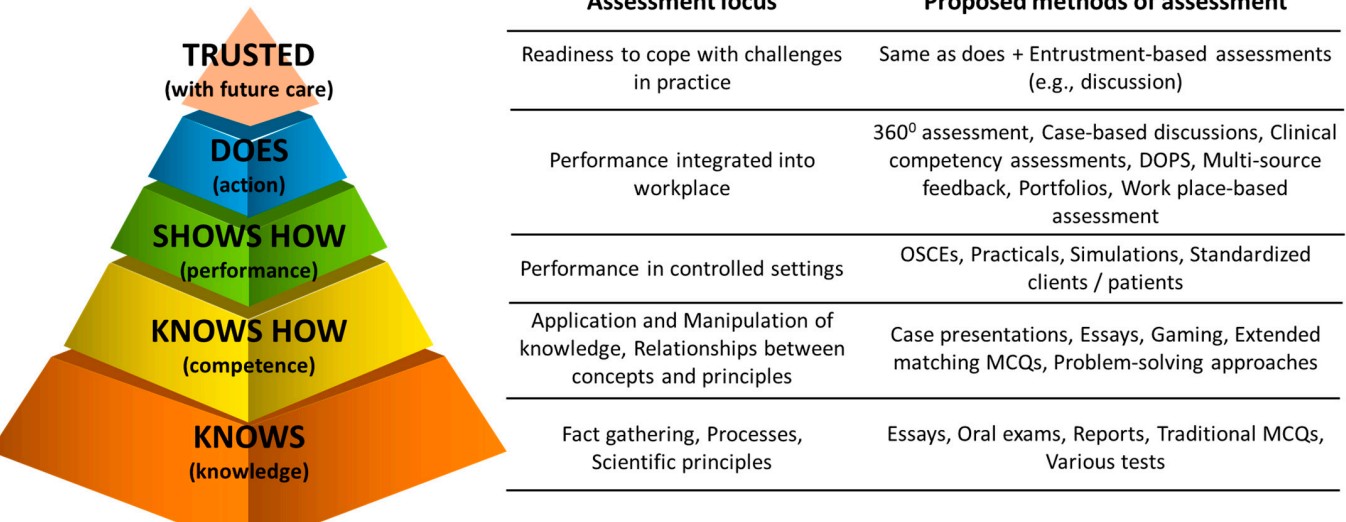

| | Assessment focus | Proposed methods of assessment |
|---|---|---|
| **TRUSTED** (with future care) | Readiness to cope with challenges in practice | Same as does + Entrustment-based assessments (e.g., discussion) |
| **DOES** (action) | Performance integrated into workplace | 360° assessment, Case-based discussions, Clinical competency assessments, DOPS, Multi-source feedback, Portfolios, Work place-based assessment |
| **SHOWS HOW** (performance) | Performance in controlled settings | OSCEs, Practicals, Simulations, Standardized clients / patients |
| **KNOWS HOW** (competence) | Application and Manipulation of knowledge, Relationships between concepts and principles | Case presentations, Essays, Gaming, Extended matching MCQs, Problem-solving approaches |
| **KNOWS** (knowledge) | Fact gathering, Processes, Scientific principles | Essays, Oral exams, Reports, Traditional MCQs, Various tests |

**Figure 1.** Extended Miller's pyramid of clinical competence of learners that can be useful to frame minds of veterinary instructors when assessing learners or planning learning and assessment activities (Modified from [13,14]). DOPS—Direct Observation of Procedural Skills. MCQ—Multi Choice Question. OSCE—Objective Structured Clinical Examination.

Overall, the frameworks for assessment of clinical competencies of learners have been classified as (1) analytical, that deconstruct clinical competence into individual pieces (e.g., attitudes, knowledge and skills), and each of these is assessed separately (e.g., ACGME framework—Accreditation Council for Graduate Medical Education); (2) synthetic, that attempt to view clinical competency comprehensively (synthesis of attitudes, knowledge and skills), assessing clinical performance in real-world activities (e.g., EPA—Entrustable Professional Activity or RIME frameworks); (3) developmental, that relates to milestones in the progression towards a clinical competence (e.g., ACGME or RIME frameworks and the novice to expert approach); or (4) hybrid, that incorporates assessment of clinical competency from a mixture of the above (e.g., CanMEDs framework, derived from the old acronym of Canadian Medical Education for Specialists) [16,18–20].

In contrast, despite a significant proportion of veterinary medical education occurring in clinical settings, the literature describing approaches to assessment of the clinical competency of learners is limited [18]. Taking into consideration the One Health approach, methodologies used in human medicine should be applicable in veterinary medicine. Medical education literature has a much larger body of evidence indicating that some of these methods work in various fields of medicine, including in-patient and out-patient care.

*1.2. Common Traps in Assessment of Clinical Competency in Veterinary Learners*

Common traps include (1) assessing theoretical knowledge rather than clinical competencies [13,14]; (2) assessing learners against each other rather than set standards [21]; (3) belief that pass/fail or similar grading system negatively affects clinical competency of learners compared to tiered grading system [22,23]; (4) different weightings for attitudes, knowledge and skills [24]; (5) leniency-bias ('fail-to-fail' or 'halo') using unstructured assessments of the clinical competency of learners [16,22,25]; and (6) occasional rather than continuous assessment of the learner's performance [24]. A plethora of assessment methods that can objectively assess the clinical competency of the learner in the medical field have emerged in the past few decades [14,16,18] but very few have been adopted in the veterinary medicine setting.

## 2. (O)RIME Framework for Assessment of Clinical Competency of Learners

One method of assessment that is used and is transferable to veterinary medicine is the (Observer)—Reporter—Interpreter—Manager—Educator ((O)RIME) framework. The RIME framework was first described by Pangaro (1999) and has since been reported in a variety of settings with a minimal need for adaptations. [19,24,26]. In essence, the RIME framework of assessment is a tool used to formatively assess a student's ability to synthesize knowledge, skills and attitude during a clinical encounter. It classifies learners into a continuum of four levels of professional development, starting as reporters, through interpreter and manager, arriving at the mastery in the learner's level of clinical competency, namely educator. A major addition to this model proposed by Battistini et al. (2002), was adding an initial step in the professional development of a learner—observer (ORIME) [21,27]—referring to the student's ability to pay attention and perceive with open-mindedness, events around them.

*2.1. Levels of Clinical Competence of Learners Using the (O)RIME Model of Assessment*

Learners show significant differences in their clinical competencies at the moment of reaching various levels of the ORIME framework. We will briefly present the typical expectations with progression of veterinary learners through the (O)RIME framework (Figure 2) [24,27–30].

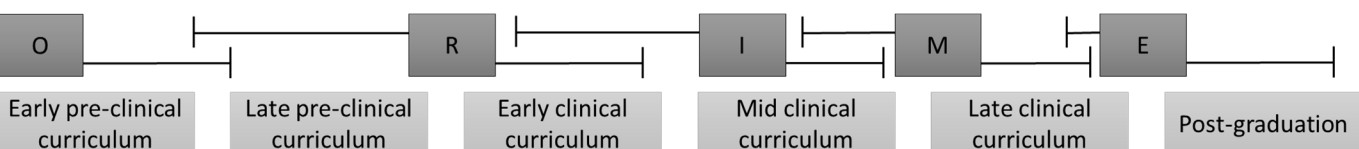

**Figure 2.** A typical expectations or progression of veterinary learners through the (O)RIME framework. A significant overlap of the times when a particular learner progresses from one level of the framework to the next is common. O—Observer. R—Reporter. I—Interpreter. M—Manager. E—Educator. Prepared using information from [24,27–29].

Learners at the **observer** level, typically representative of the very early part of the curriculum, lack skills to conduct a comprehensive health interview and/or present the clinical encounter at rounds or to peers or instructors. Learners at the observer level lack competencies that would contribute to the management of the particular case and patient care.

Learners at the **reporter** level, typically representative of the late part of the pre-clinical and early part of the clinical curriculum, should be capable of gathering reliable clinical information, preparation of basic clinical notes, able to differentiate normal from abnormal, and present their findings to peers and/or instructors.

Learners at the **interpreter** level, typically representative of the early to mid-part of the clinical curriculum, should be capable of organizing gathered information in a logical way, preparing a prioritized list of differential diagnoses without prodding, and able to

support their arguments for inclusion/exclusion of particular diagnoses/tests. They may or may not be able to propose a management plan for the clinical encounter.

Learners at the **manager** level, typically representative of the mid- to late-part of the clinical curriculum, should be capable of summarizing the gathered information in a logical way using veterinary medical language, preparing a prioritized list of differential diagnoses, support their arguments and propose an appropriate management plan. At the manager level, the learner should consider the client's circumstances, needs and preferences.

Finally, learners at the **educator** level, typically representative of the advanced part of the clinical curriculum, should be capable of doing all of the above coupled with a critique of the encounter, including important omissions and further research questions, and present the case in a way that can educate others. Learners at this level are truly self-directed. It is a reality that some learners do not reach the level of educator by the time they graduate from the veterinary school and may need 2–3 years post-graduation or residency to reach this level.

A significant overlap in the levels of achieved clinical competency is common [24,27,29]. This, in part, may be a result of the prior exposure to and complexity of the clinical encounter the learner is dealing with. For example, a learner in familiar or non-complicated clinical encounters may work at Manager level, but for the complicated or unfamiliar clinical encounters may operate at Interpreter level. Indeed, a very complicated clinical encounter may even result in the learner, operating at Reporter level.

## 2.2. Advantages of the (O)RIME Framework

The major advantages of the (O)RIME framework include

- A systematic structure of expectations that can be coupled with day one competencies to guide students though their education, and allow Faculty to assess and evaluate the introduction, implementation and assessment of these skills in the curriculum [30].
- Allows for early detection of at-risk learners [17,21,28,31]. These identified learners need immediate attention. The prompt addressing of poor performance has been stated as a priority for any medical, presumably including veterinary medical, education delivered in an outcome-based training mode [32].
- Allows for standardized assessment of clinical competencies of learners and, when implemented correctly, should prevent assessment of learners relative to each-other [21].
- As the narrative associated with the (O)RIME framework is easy to understand, it benefits the learner both during feedback sessions or at receiving a report card [26,33].
- At each level, it assesses a synthesis of attitudes, knowledge and skills, rather than assessing them individually as done with many other frameworks of assessment.
- Can be used for assessment of a single clinical encounter, day, week or an entire course. Hence, the (O)RIME framework can be used as an assessment method for the creating of a regular record of in-training assessment (RITA) [34]
- Proven to possess good repeatability and reliability in a variety of human medical clinical settings [18,26,28].
- The ease of relating assessment to Bloom's taxonomy of learning [35,36], as the five levels of competency represent a developmental framework of progressively higher cognitive skills achieved by the learner; that is, from data gathering to analysis, synthesis, and evaluation [28].
- The framework takes advantage of a clinician's ability to draw conclusions from observations and data, and uses that the same diagnostic approach in the assessment of students whilst also addressing the emotional difficulty teachers have in "giving" a grade, e.g., "does what I see before me in this patient fit better with xxx" translates into "does what I see before me in this student fit better with reporter, interpreter or manager?" [37].

*2.3. Limitations of the (O)RIME Framework*

(O)RIME framework has some limitations.

- The meaning of words and terminology are not always self-evident, e.g., "reporting" is not simply repeating the facts but is about the process of getting the facts [37].
- The (O)RIME framework is not suitable for assessing individual skills. It rather assesses the overall clinical competency of the learner [21], albeit potentially within a single clinical encounter. This is completely understandable as the framework is a synthetic method of assessment, not analytical.
- During a single clinical encounter a learner may demonstrate capacity from several domains of the (O)RIME framework [33]. This may occur in any of the assessments. Analytical methods of assessment may suffer less from this problem.
- The framework is not designed for assessment of non-technical skills and competencies but rather for only technical skills and competencies [26,33]. Hence, Holmes et al. (2014) recommended the use of the (P)RIME framework, where P is for professionalism [33]. An alternative assessment method for soft skills is the multi-sourced feedback [34]. However, the suggested limitation of the framework related to the assessment of non-technical skills may not be entirely true, as some studies have shown it to be effective in assessing these skills and competencies of learners [38]. Clearly, this 'limitation' requires further investigation.
- The framework is often not recommended as a sole method of assessment of the progression of learners [21], particularly at schools relying on a tier mark-associated grading system. Hence, some authors have recommended it to be used only as part of the toolbox of assessment of the progression of learners [21] coupled with methodologies such as direct observations.
- For fulfilling the assessment requirements, the (O)RIME framework requires team involvement (observations by all team members), rather than grading by a single person. Hence, some organizational skills are required to ensure 'a round table discussion' occurs before the level is discussed by a nominated person with the learner [39]. However, this limitation of the framework may not be always true as some studies have shown good assessment characteristics using the learners' progression judged by single assessors [40].
- Educators can get confused that the (O)RIME is a developmental framework in which students go through the phases. It is not. When the student moves from 'reporter' to 'interpreter', they do not stop being a reporter. When they move to 'manager', they must continue to gather information and interpret it.

## 3. Examples of Bovine Clinical Encounters

We provide two examples of the use of the (O)RIME method both in assessment of clinical reasoning competencies and provision of effective feedback using two bovine clinical encounters. From the examples, readers should be able to extract the basics of the method and start using it in their day-to-day practice. The intent is that this review will facilitate discussion and/or research in the areas of assessment of clinical reasoning and effective feedback in clinical teaching of learners of veterinary medicine.

*3.1. Example Clinical Encounter: Left Displaced Abomasum in a Dairy Cow*

As the first example clinical encounter, we used a case of left displaced abomasum in a mature dairy cow. The ORIME model for the clinical encounter is presented in Table 1. Feedback must be effective and stimulate further development of deep learning and progression in clinical competencies of the leaner [21].

**Table 1.** Example clinical encounter: Left displaced abomasum in a mature dairy cow.

| Learner Level | Descriptor | Learner Report | Effective Instructor's Feedback [1] |
|---|---|---|---|
| Observer | Bystander. Merely describing data | 'We were presented with a Holstein Friesian cow that calved 3 weeks ago. The client reported sudden loss of appetite and a drop in milk production was detected yesterday afternoon and is still present.' | 'This is a really good start. Do you know the age of the cow? This would be very helpful. Based on the information you have gathered, how you would examine this patient if I was not around?' |
| Reporter | Understanding 'what is wrong' and some of the 'why it is wrong' | 'We were presented with a 5-yr old, Holstein Friesian cow that calved 3 weeks ago. The client reported a sudden inappetence and a drop in milk production from last night. Major abnormalities detected on clinical examination included 4+ ketone bodies in urine, ketone odor of the breath, rumen fill of score 1, rumen atony, left side 'ping' in the area of the last 2 ribs on the left side, and sloshing sounds on succussion.' | 'This is a really good summary. Now, can you think of some reasons for these detected signs?' 'This is a really good summary. Now, based on the detected signs, it will be great to come up with a list of 3–4 differential diagnoses that should be considered.' |
| Interpreter | Understanding 'why it is wrong' | ' . . . based on the presenting signs, I believe she suffers from left displaced abomasum with a secondary ketosis due to the loss of appetite. Another possibility that should be considered is a LDA secondary to metritis. However, the basic clinical examination gave no indications of metritis.' | 'Excellent differential diagnosis. You mentioned metritis. I would be very interested to hear your reasoning that resulted in the elimination of metritis as a primary diagnosis in this case.' 'Excellent differential diagnosis. You mentioned secondary ketosis. I would be interested to hear your reasoning as why it has occurred in this case' |
| Manager | Understanding 'how to address the problem' demonstrating prioritization and skills of analysis of major problems. Considers client's particular circumstances. | ' . . . I would like to ensure that she really has LDA before opting for surgery. As the cow is in the clinic, options that I consider in the workup include ultrasonography and/or Liptak test. If the diagnosis is confirmed, LDA surgery by right side approach will be recommended. Post-operatively, I would consider giving the cow some propylene glycol and electrolytes by oral administration. Surgical correction should result in full recovery. As the transit of ingesta return, ketosis should self-cure.' | 'This sounds like an excellent approach to reach the final diagnosis and very well-planned management of the case. Please elaborate the reason you considered this surgical approach for this particular case (NOTE: e.g., Client has experience and/or preference), and the post-operative care instructions to the client.' I like your approach. I would be interested to hear what you would be looking for on ultrasound to confirm your diagnosis and surgical approach' |
| Educator | Commit to self-learning, demonstration of reflection, and education of the team | ' . . . This clinical encounter meets the criteria for a primary LDA with a secondary ketosis. Research indicates that surgical correction of LDA results in self-cure of ketosis but return to full production and good fertility indicators benefit from 3–7 days propylene glycol treatment. I discussed with the client the importance of the transition cow diet and its role in prevention of LDA in the future.' | 'Well done. Specifically, you managed this clinical encounter well and educated the client on the reasons for LDA occurring. I think you are ready for more complicated clinical encounters' |

Prepared using information presented in [6,21,24,29,41,42]. [1] Feedback uses elements of the 5 microskills model of clinical teaching [6,43–45].

*3.2. Example Clinical Encounter: Left Displaced Abomasum in a Dairy Cow*

As the second example clinical encounter, we used a case of apparently anestric mature dairy cow. The ORIME model for the clinical encounter is presented in Table 2.

**Table 2.** Example clinical encounter: Apparently anestrus mature dairy cow.

| Learner Level | Descriptor | Learner Report | Effective Instructor's Feedback [1] |
|---|---|---|---|
| Observer | Bystander. Merely describing data | 'We were presented with a Jersey cow that calved 3 months ago. The client reported sudden loss of appetite and a drop in milk production was detected yesterday afternoon and is still present.' | 'This is a really good start. Do you know the age of the cow? This would be very helpful. Based on the information you have gathered, how you would examine this patient if I was not around?' |
| Reporter | Understanding 'what is wrong' and some of the 'why it is wrong' | 'We were presented with a 5-yr old, Holstein Friesian cow that calved 3 weeks ago. The client reported a sudden inappetence and a drop in milk production from last night. Major abnormalities detected on clinical examination included 4+ ketone bodies in urine, ketone odor of the breath, rumen fill of score 1, rumen atony, left side 'ping' in the area of the last 2 ribs on the left side, and sloshing sounds on succussion.' | 'This is a really good summary. Now, can you think of some reasons for these detected signs?' 'This is a really good summary. Now, based on the detected signs, it will be great to come up with a list of 3–4 differential diagnoses that should be considered.' |
| Interpreter | Understanding 'why it is wrong' | ' … based on the presenting signs, I believe she suffers from left displaced abomasum with a secondary ketosis due to the loss of appetite. Another possibility that should be considered is a LDA secondary to metritis. However, the basic clinical examination gave no indications of metritis.' | 'Excellent differential diagnosis. You mentioned metritis. I would be very interested to hear your reasoning that resulted in the elimination of metritis as a primary diagnosis in this case.' 'Excellent differential diagnosis. You mentioned secondary ketosis. I would be interested to hear your reasoning as why it has occurred in this case' |
| Manager | Understanding 'how to address the problem' demonstrating prioritization and skills of analysis of major problems. Considers client's particular circumstances. | ' … I would like to ensure that she really has LDA before opting for surgery. As the cow is in the clinic, options that I consider in the workup include ultrasonography and/or Liptak test. If the diagnosis is confirmed, LDA surgery by right side approach will be recommended. Post-operatively, I would consider giving the cow some propylene glycol and electrolytes by oral administration. Surgical correction should result in full recovery. As the transit of ingesta return, ketosis should self-cure.' | 'This sounds like an excellent approach to reach the final diagnosis and very well-planned management of the case. Please elaborate the reason you considered this surgical approach for this particular case (NOTE: e.g., Client has experience and/or preference), and the post-operative care instructions to the client.' I like your approach. I would be interested to hear what you would be looking for on ultrasound to confirm your diagnosis and surgical approach' |
| Educator | Commit to self-learning, demonstration of reflection, and education of the team | ' … This clinical encounter meets the criteria for a primary LDA with a secondary ketosis. Research indicates that surgical correction of LDA results in self-cure of ketosis but return to full production and good fertility indicators benefit from 3–7 days propylene glycol treatment. I discussed with the client the importance of the transition cow diet and its role in prevention of LDA in the future.' | 'Well done. Specifically, you managed this clinical encounter well and educated the client on the reasons for LDA occurring. I think you are ready for more complicated clinical encounters' |

Prepared using information presented in [6,21,24,29,41,42]. [1] Feedback uses elements of the 5 microskills model of clinical teaching [6,43–45].

## 4. Discussion

The primary aim of preparing this review was to describe a teaching technique used in other disciplines of the OneHealth initiative that is not currently used in the teaching of veterinary medicine, but has been identified as a useful technique based on the success of its use in other disciplines. Assessment of the clinical competency of veterinary learners is a major and, often, challenging task for many veterinary schools. In fact, many of the assessment frameworks lack reliability. For example, a reliable assessment using the multi-source feedback (used by a few schools that we are aware of), requires a minimum of seven assessment forms. In reality, seven available forms are rarely, if ever, reached [24], which compromises the credibility of the assessment of the clinical competency of veterinary

learners using that tool. Other assessment methods have good reliability [46] but are not easily implemented due to high work load, unwillingness to participate, or lack of feedback information to the learner (e.g., script concordance questions).

It is our view that the (O)RIME framework has the potential to be an attractive assessment method for measuring and reporting the progression of clinical competencies in veterinary learners; however, it is not suitable for assessment of individual skills (not being subject of this review (NOTE: Readers are urged to seek suitable literature on assessment of individual skills.). The framework has been tested and proven to be useful in a variety of setting and branches of human medical education [19,24,26,29,30,42,47–50]. Some studies have assessed usefulness in assessment of multi-clerk medical educational facilities [29,42]. Interestingly, single assessor descriptors of progression using the (O)RIME framework have been compared and correlated well with other final scoring systems assessing performance of learners during their clinical rotations [40]. Additionally, the framework has been proven to correlate well between non-trained experienced and less experienced instructors and residents [40]. This is an important finding as instructor experience and training are highly influential in the grading of many other assessment frameworks, but should be taken cautiously as it is based on the experience of in-patients in a single institution. Indeed, training in assessment using the framework resulted in even further improvements of performance [40].

Although results from studies in the field of medical education provide some comfort, for a strong recommendation for use of the (O)RIME framework to be made, research in the field of veterinary medical education is required. The research should test the assessment characteristics of the framework but also its acceptability, educational effect (see below) and feasibility [24,25,51].

As a minimum, the framework needs to be tested for its reliability and validity. **Reliability** of an assessment method (some authors prefer the term generalizability) describes the extent to which a particular score of the measurement of the clinical competency of the learner would yield the same results on repeated assessment (i.e., what is the reproducibility of scores obtained by that particular method) [51,52]. For the framework to be accepted as an attractive assessment method for measuring clinical competence of veterinary learners, reliability should be at least 80% [24]. However, some studies in human medicine have found the framework performing overall at lesser values (e.g., as low as ~75% [42]) in part due to the subspecialty studied and the need for insufficient assessments. However, this is yet higher than for other proposed assessment tasks, such as the IDEA (interpretive summary, differential diagnosis, explanation of reasoning, and alternatives) that overall had just over 50% reliability [37,53]. **Validity** of an assessment method describes the degree to which the inferences made about the clinical competence of the learner are correct (i.e., does the method measure what it is supposed to do?) [51,52].

Implementation of the (O)RIME framework should not be based only on quantitative information regarding test performance. The test should be acceptable, educationally suitable and feasible for the institution (or clinical workplace) carrying out the assessment **Acceptability** of an assessment method describes the extent to which stakeholders endorse the method of assessing the clinical competency of a learner using that particular assessment method [51,52]. The framework has been reported as acceptable for institutions, instructors and learners [30]. **Educational effect** of an assessment method describes the effect on learner's motivation to improve their clinical competency [51,52]. **Feasibility** of an assessment method describes the degree to which the assessment method is affordable and efficient in testing clinical competency of learners [24,51,52]. Introduction of any new assessment method may be seen as additional work and willingness may be lacking in learners and also instructors [24]. Fortunately, the implementation of the (O)RIME framework has been judged positively by medical learners in a few fields [25,49] for the ease of understanding and the feedback.

*Usefullness of (O)RIME for Veterinary Medical Education*

**Adjusting the level of clinical encounters to meet learners needs.** Although this is not always possible, the (O)RIME framework provides an early indication of a learner's level and so (if possible) clinical encounters may be adjusted to best suit the learner's needs (e.g., less or more complicated cases). For example, clinical encounters involving multiple animals and multiple problems provide a level of complication that extends the learners' clinical reasoning beyond the individual patient with a single problem, i.e., a simple case of LDA in a dairy herd may be very challenging for one learner, but less challenging for another. Providing the opportunity to challenge the learner by asking them to investigate multiple LDAs occurring in a dairy herd due to dystocia adjusts the level of learning to suit their needs, whilst not negatively effecting the needs of the other learner.

**Detection of at-risk learner**. When a learner is detected to be lagging behind the expected standard (rather than behind their peers), an immediate and clear feedback on the expected standards required for that particular level of the curriculum (see Figure 2) should be given [24]. Although the feedback may need to be delayed if it was based on a single clinical encounter, particularly if the encounter had been very complicated and/or unfamiliar to the learner. A note should be made to follow the particular learner and if a similar performance is seen in another clinical encounter, a feedback session should be scheduled. For example, a final year veterinary student performing at the level of Reporter (summarizes the information well but is unable to provide differential diagnoses) rather than Manager, in multiple clinical encounters of varying degrees of difficulty and complication requires feedback to assist them in achieving the expected standard.

**Documentation of deficiency in core expected outcomes**. Regular use of the framework may indicate that a larger proportion of learners (e.g., all or a particular cohort) lack core expected outcomes. Provided this is detected early, remedial action may be taken for the existing cohort or, at least, for the subsequent cohorts of learners. For example, student's being successful at one "level" requires the student to have the right combination of knowledge, skills and attitude to address the clinical encounter. A lack of ability to 'shift' to the next level that is identified in multiple learners may indicate a curriculum deficiency which can be addressed before the end of the student's formal education.

**Improving communication between faculty, institutions, instructors and learners**. As regular feedback is a feature of the framework, this assessment method can be used to improve communication channels between the stakeholders [30].

## 5. Conclusions

Preparing graduates to meet day one veterinary graduate attributes requires effective clinical teaching techniques coupled with effective feedback. Consistency of assessment of clinical reasoning is difficult in the variety of settings utilized to educate veterinary learners. The ORIME framework is a developmental approach that distinguishes between basic and advanced levels of performance. Each step represents a synthesis of skills, knowledge and attitude. The ORIME framework provides assessors with a clear, minimal level descriptive required for each level of training rather than a numerical grading system. We expect this framework will be acceptable for ensuring graduates meet essential Day One competencies and should be suitable for veterinary schools' accreditors and stakeholders involved in veterinary education and veterinary profession. Within veterinary medicine education, the use of frameworks like ORIME for assessment and feedback are lacking. Research into the effectiveness of ORIME in the medical profession is abundant and it has been found to be a practical, useful assessment and feedback tool that is highly acceptable to learners and instructors. It is hoped that this review stimulates discussion and research into this important area.

## 6. Glossary of Terms

**Clinical competency**—ability to consistently select and carry out relevant clinical tasks pertaining to the particular clinical encounter in order to resolve the health or productivity

problems for the client, industry and patient, in an economic, effective, efficient and humane manner, followed by self-reflection on the performance.

**Clinical encounter**—any physical or virtual contact with a veterinary patient and client (e.g., owner, employee of an enterprise) with a primary responsibility to carry out clinical assessment or activity.

**Clinical instructor**—in addition to the regular veterinary practitioner's duties, a clinical instructor should fulfil roles of assessor, facilitator, mentor, preceptor, role-model, supervisor, and teacher of veterinary learners in a clinical teaching environment. Apprentice/intern in the upper years, Resident, Veterinary educator/teacher, Veterinary practitioner.

**Clinical reasoning**—process during which a learner collects information, process it, comes to an understanding of the problem presented during a clinical encounter, and prepares a management plan, followed by evaluation of the outcome and self-reflection. Common synonyms include clinical acumen, clinical critical thinking, clinical decision-making, clinical judgment, clinical problem-solving, and clinical rationale.

**Clinical teaching**—form of an interpersonal communication between a clinical instructor and a learner that involves a physical or virtual clinical encounter.

**Deep learning**—aiming for mastery of essential academic content; thinking critically and solving complex problems; working collaboratively and communicating effectively; having an academic mindset; and being empowered through self-directed learning.

**Effective feedback**—purposeful conversation between the learner and instructor within a context and a culture with aim of stimulating self-reflection as a powerful tool for deep learning.

**Self-directed learning**—learners take charge of their own learning process by identifying learning needs, goals, and strategies and evaluating learning performances and outcomes. Learner-centered approach to learning.

**Work-based learning**—educational method that immerses the learners in the workplace.

**Author Contributions:** All authors have contributed to this work. All authors have read and agreed to the published version of the manuscript.

**Funding:** This research received no external funding.

**Institutional Review Board Statement:** Not applicable.

**Informed Consent Statement:** Not applicable.

**Data Availability Statement:** Not applicable.

**Conflicts of Interest:** The authors declare no conflict of interest.

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
