# Peer review of "Practical Use of the (Observer)—Reporter—Interpreter—Manager—Expert ((O)RIME) Framework in Veterinary Clinical Teaching with a Clinical Example"

_encyclopedia, doi:10.3390/encyclopedia2040113_

Round 1

Reviewer 1 Report

The manuscript presented key concepts in health curriculum, in this case, the veterinary curriculum and assessment of day-one competencies. It appears generalizable to the clinical competencies but fits less into the preclinical and paraclinical assessment areas. Specific queries and comments have been inserted in the attached document for consideration by the authors. Adherence should improve the outcome. The authors also need to ensure that their primary reason for carrying out this study is well delivered in the discussion section. It looks a bit too generic at the moment.

Author Response

Thanks to the reviewer for the constructive criticism.  We have addressed all reviewer’s comments (red font).

The manuscript presented key concepts in health curriculum, in this case, the veterinary curriculum and assessment of day-one competencies. It appears generalizable to the clinical competencies but fits less into the preclinical and paraclinical assessment areas. Specific queries and comments have been inserted in the attached document for consideration by the authors. Adherence should improve the outcome. The authors also need to ensure that their primary reason for carrying out this study is well delivered in the discussion section. It looks a bit too generic at the moment.

The primary reason for preparing this review was to describe a teaching technique used in other disciplines that is not currently used in the teaching of veterinary medicine, but has been identified as a useful technique based on the success of its use in other disciplines. (Discussion, Lines 2-5)

PDF comments

L19 - Rather use 'Contrastingly'. By using 'unfortunately', it means that you have for and against the subject of investigation, and it will lead to courtesy bias in investigation. Adjusted (L19/20)

L22 - that the Adjusted (L22)

L23 - Did you propose anything at the end of the study? If yes, mention it. Adjusted (L24-25)

L28 - new professional entrants Adjusted (L22)

L30 - Is 'competencies' not a better term? This is what is proposed by the Education Committee of OIE. Adjusted (L30-32)

L50 - Add space here. Adjusted (L52)

L69 - In contrast, - 'Unfortunately' is a subjective term in science. Adjusted (L71)

L75 – 77 This paragraph should be moved before section 2, which explained ORIME. That is where it fits best and flow. Adjusted (L92-93)

L212-216 Remove this last bullet. Adjusted (L220)

L225 Instead of 'We will use', it should read 'we used' since the examples was already used. Adjusted (L229 and again Section 3.2 L2)

Table 1 - If there are additional notes or documentation that assisted in the compilation of this table, it should be presented as supplementary materials- and the table should not just be an hyper-summary. No additional material available.

Discussion L5 - What are these seven forms? Do not assume that all readers will know. This journal is not for the academia in the space of curriculum evaluation alone. Adjusted (Discussion L9)

Discussion L44 - Use 'quantitative' instead of 'statistical'. Adjusted (Discussion Line 50/51)

Discussion L103 - There need to be some thoughts on the general acceptability of the tool among the relevant stakeholders, the professional councils, the students, the industry etc to know if it meets the needs of the day-one competencies expected of a recent graduate. Adjusted (Discussion L105-107)

Discussion L104 – of terms Adjusted (Discussion Line 113)

Reviewer 2 Report

This review well explores the use of the model (RIME) in evaluating clinical reasoning skills to be applied to veterinary students or as feedback on their professional progress. It does not specifically address the practical individual skills as well as the theoretical ones that are the basis for adopting a diagnosis or clinical reasoning. . It would be desirable to insert how to evaluate the theoretical preparation of the student and the possible methods of delivery of the same by the teacher which should be as uniform as possible

Author Response

Thanks to the reviewer for the constructive criticism.  We have addressed all reviewer’s comments (red font).

This review well explores the use of the model (RIME) in evaluating clinical reasoning skills to be applied to veterinary students or as feedback on their professional progress. It does not specifically address the practical individual skills as well as the theoretical ones that are the basis for adopting a diagnosis or clinical reasoning.

The (O)RIME model is not suitable for assessment of individual skills (as stated in Section 2.3. Adjusted (Discussion Lines 17-18)

It would be desirable to insert how to evaluate the theoretical preparation of the student and the possible methods of delivery of the same by the teacher which should be as uniform as possible

Adjusted (Discussion Lines 18-19)

Round 2

Reviewer 1 Report

No further comment. The manuscript is recommended for acceptance.

Reviewer 2 Report

I have no comments